# Cyclooxygenase and Lipoxygenase Gene Expression in the Inflammogenesis of Colorectal Cancer: Correlated Expression of EGFR, JAK STAT and Src Genes, and a Natural Antisense Transcript, RP11-C67.2.2

**DOI:** 10.3390/cancers15082380

**Published:** 2023-04-20

**Authors:** Brian M. Kennedy, Randall E. Harris

**Affiliations:** Colleges of Public Health and Medicine, The Ohio State University Comprehensive Cancer Center, The Ohio State University, 1841 Neil Avenue, Columbus, OH 43210-1351, USA

**Keywords:** colorectal cancer, cyclooxygenase, lipoxygenase, mRNA sequence data, EGFR, JAK STAT, Src, natural antisense transcript, RP11-67C2.2

## Abstract

**Simple Summary:**

Since chronic inflammation has been found to increase the risk of many forms of cancer, we examined the expression of major genes that regulate inflammation in cancerous specimens from 469 patients with colorectal cancer. We found that these genes were overexpressed together with several other cancer-promoting genes in colorectal tumors. Our findings demonstrate that the development of colorectal cancer is tightly linked to the overexpression of genes that regulate inflammation. These genes and their products are molecular targets for cancer prevention and therapy.

**Abstract:**

We examined the expression of major inflammatory genes, cyclooxygenase-1, 2 (COX1, COX2), arachidonate-5-lipoxygenase (ALOX5), and arachidonate-5-lipoxygenase activating protein (ALOX5AP) among 469 tumor specimens of colorectal cancer in The Cancer Genome Atlas (TCGA). Among 411 specimens without mutations in mismatch repair (MMR) genes, the mean expression of each of the inflammatory genes ranked above the 80th percentile, and the overall mean cyclooxygenase expression (COX1+COX2) ranked in the upper 99th percentile of all genes. Similar levels were observed for 58 cases with MMR mutations. Pearson correlation coefficients exceeding r = 0.70 were observed between COX and LOX mRNA levels with genes of major cell-signaling pathways involved in tumorigenesis (Src, JAK STAT, MAPK, PI3K). We observed a novel association (r = 0.78) between ALOX5 expression and a natural antisense transcript (NAT), RP11-67C2.2, a long non-coding mRNA gene, 462 base pairs in length that is located within the terminal intron of the ALOX5 gene on chromosome 10q11.21. Tumor-promoting genes highly correlated with the expression of COX1, COX2, ALOX5 and ALOX5AP are known to increase mitogenesis, mutagenesis, angiogenesis, cell survival, immunosuppression and metastasis in the inflammogenesis of colorectal cancer. These genes and the novel NAT, RP1167C2.2 are potential molecular targets for chemoprevention and therapy of colorectal cancer.

## 1. Introduction

Inflammation is an essential feature of innate immunity in human physiology. The process, ordinarily under tight regulation, has been designed by nature to identify, confine, and destroy noxious elements that enter the body. Signaling molecules called eicosanoids modulate the human inflammatory response. These molecules are synthesized from long chain fatty acids, principally arachidonic acid, that are abundant in cell membranes. Two major pathways regulate inflammation, the cyclooxygenase (COX) pathway that is regulated by prostaglandins in the E series such as PGE2, and the lipoxygenase (LOX) pathway that is regulated by leukotrienes such as leukotriene B4 and cysteinyl leukotrienes. The COX pathway is inhibited by aspirin and other nonsteroidal anti-inflammatory drugs whereas LOX is inhibited by antihistamines and other allergic medications. Sustained hyperactivity of either of these pathways can propel the pathogenesis of chronic inflammatory conditions, e.g., overexpression of COX in rheumatoid arthritis and inflammatory bowel disease, and overexpression of LOX in allergic rhinitis and bronchial asthma [1,2,3,4].

The enzyme, cyclooxygenase, activates the COX inflammatory cascade by catalyzing the biosynthesis of prostaglandins from arachidonic acid. Two primary genes encode cyclooxygenase, a constitutive gene (COX1) located at 9q32, and its inducible isoform (COX2) located at 1q25 [5,6,7].

Cyclooxygenase-1 (COX1) is widely expressed in most tissues and can be thought of as a “housekeeping enzyme”. Eicosanoids derived from COX1 include prostaglandin E2 (PGE2) and prostacyclin (PGI2) that are cytoprotective in the stomach, and thromboxane (TxA2) that stimulates platelet aggregation and vasoconstriction in blood vessels [8].

Cyclooxygenase-2 (COX2) is an inducible gene that is normally silent in the absence of inflammatory stimuli. The induction of COX2 can be triggered by a variety of inflammatory factors (e.g., tobacco, alcohol, ischemia, trauma, pressure, foreign bodies, toxins, bacteria, viruses, and lipopolysaccharides). Induction of the gene quickly results in cyclooxygenase-catalyzed biosynthesis of prostaglandins of the E-series, particularly prostaglandin E2 (PGE2), that orchestrate the inflammatory response. Molecular and epidemiologic studies provide consistent evidence that sustained overexpression of COX2 is a major factor in the development of many malignancies including colorectal cancer [9,10,11,12].

Arachidonic acid is the major substrate for cyclooxygenase and lipoxygenase activity and the biosynthesis of prostaglandins and leukotrienes. This polyunsaturated omega-6 fatty acid is derived from the hydrolysis of phospholipids in cell membranes catalyzed by enzymes of the cytosolic phospholipase A2 family. Notably, one gene encoding the phospholipase enzyme, PLA2G4A, is tightly linked to the COX2 gene on chromosome 1 [13].

Leukotrienes are a family of eicosanoid mediators of the LOX pathway that are primarily responsible for allergic inflammatory conditions. For example, Leukotriene B4 (LTB4) stimulates neutrophil chemotaxis, activation and degranulation in conditions such as allergic rhinitis, and cysteinyl leukotrienes (CysLT) induce bronchoconstriction, eosinophilia and edema associated with bronchial asthma. These molecules are synthesized in leukocytes and other immune cells by the oxidation of arachidonic acid and the essential fatty acid, eicosapentaenoic acid. Two enzymes catalyze leukotriene biosynthesis: arachidonate 5-lipoxygenase (encoded by the ALOX5 gene) and arachidonate 5-lipoxygenase activating protein (ALOX5AP), also called FLAP (encoded by the ALOX5AP gene). The ALOX5AP protein binds and activates lipoxygenase at the cell membrane and strictly regulates the activity of ALOX5. The ALOX5 gene is located at 10q11.21 and ALOX5AP is at 13q12.3 [3,14]. In addition to the genesis of allergic conditions, leukotrienes have also been implicated in the initiation and progression of atherosclerosis as well as carcinogenesis and tumor progression [15,16].

More than 150 years ago, the German pathologist, Rudolf Virchow, suggested that chronic inflammation leads to cancer development by increasing cellular proliferation [17,18,19]. In recent years, a cohesive body of scientific evidence has confirmed the validity of Virchow’s original hypothesis, and various models of carcinogenesis have been proposed involving inflammatory stimuli of the COX and LOX cascades [9,11,20,21].

Several independent investigators have observed that there is dual involvement of the COX and LOX cascades in the development of multiple forms of cancer [16,22,23], and reciprocally, compounds such as curcumin and licofelone with activity against both inflammatory pathways have shown significant anti-cancer effects in cell lines and animal models of breast cancer, colon cancer, prostate cancer, and lung cancer [24,25,26,27]. In a recent case–control study, the combined use of COX and LOX inhibitors was found to be associated with a greater reduction in human breast cancer risk than individual agents per se [28].

In addition to inflammatory genes, certain alleles of the carcinoembryonic antigen-related cell adhesion (CEACAM) superfamily of genes are routinely used to monitor colon cancer progression and response to therapy. Specifically, elevated levels of CEACAM1, CEACAM5 and CEACAM6 are known to inhibit cell adhesion, differentiation and apoptosis in association with metastatic spread of cancer cells, and elevated levels are therefore predictive of recurrence and poor survival in patients with colon cancer [29].

Hereditary colorectal cancer is characterized by mutations and loss of function in a family of genes that regulate mismatch repair of DNA during cell division. Individuals with mutated alleles of certain mismatch repair genes (MLH1, MSH2, MSH6, PMS2 and EPCAM) are predisposed to the development of colorectal cancer, a genetic entity known as the Lynch Syndrome [30,31]. Cases of colorectal cancer in TCGA have been classified into those with and without mutated MMR genes by immunohistochemistry [32].

In the current study, we examined the expression of COX1, COX2, ALOX5, ALOX5AP and related genes in surgical specimens of colorectal cancer ascertained from The Cancer Genome Atlas (TCGA). We focused specifically on genes highly correlated with inflammatory genes as well as CEACAM and MMR genes. Our objective was to gain a better understanding of the role of the COX and LOX inflammatory cascades in colorectal carcinogenesis.

## 2. Materials and Methods

We analyzed gene expression data for 469 specimens of colorectal cancer generated by The Cancer Genome Atlas managed by the US National Cancer Institute and the National Human Genome Research Institute (http://cancergenome.nih.gov, accessed on 10 November 2022). Data analyzed were accessed from the open access tier of TCGA. The open access tier of TCGA contains data that cannot be attributed to an individual research participant.

Genetic expression data were generated on an Illumina HiSeq 2000 RNA sequencing platform and made available through The Cancer Genome Atlas (TCGA) of the US National Cancer Institute (NCI) and the National Human Genome Research Institute (NHGRI) (http://genome.nih.gov, accessed on 10 November 2022). Gene-level reads were first transformed to maximum likelihood expression counts by an expectation-maximization algorithm (mRNA-Seq by expectation-maximization, RSEM) and then transformed to binary logarithms, log2(x + 1), in order to approximate normality of distributions [33].

Our analysis was designed to characterize the expression of PTGS1, PTGS2 (COX1, COX2), ALOX5, ALOX5AP, and related genes for the entire database, by the presence/absence of mutations in mismatch repair genes (MMR), and by gender. Paired samples of normal colonic epithelium were also available for 45 of the cases for comparison with cancerous tissues.

We examined kernel density (violin) plots of genetic expression and conducted tests of significance of mean differences in expression by MMR status and by gender for COX1, COX2, ALOX5, ALOX5AP and related genes. Violin plots provide a mirror image of the kernel density for a given variable. Each plot shows the raw data points, the mean, standard deviation, and a smoothed density curve of the distribution.

We also estimated Pearson correlation coefficients of ALOX5, ALOX5AP, COX1, and COX2 in order to identify significant associations among these inflammatory genes and co-expression with other genes. Our analysis primarily focused on correlations with r values of 0.70 or higher corresponding to an r^2^ value of 50% or more in shared variance and with *p* < 0.001. One specific aim was to identify clusters of high correlations with genes of well-known tyrosine kinase pathways of colorectal carcinogenesis such as Janus Kinase Signal Transducers and Activators of Transcription (JAK STAT), Src Proto-oncogene Tyrosine Kinase (Src), Mitogen-Activated Tyrosine Kinase (MAPK), and the Phosphinositide 3-Kinase (PI3K) pathways [34,35,36,37,38,39,40]. For selected genes with high relevance to colon carcinogenesis, e.g., Epidermal Growth Factor Receptor (EGFR) [41,42,43], we utilized multivariate linear regression to evaluate associations with the inflammatory genes.

Expression levels of genes that were highly correlated with COX1, COX2, ALOX5 and ALOX5AP were also tested for mean differences by gender and MMR status and correlation matrices were examined to elucidate clustering of genes involved in the inflammogenesis of colorectal cancer. We used the “Bonferroni” method of conducting protected tests of significance for tests of means and correlations by dividing *p* values by 20,500. Statistical analyses were conducted using the SAS/STAT software (version 9.1.3) (SAS Institute, Inc.) and the R software package (v2.6) from Bioconductor (http://www.bioconductor.org, accessed on 10 November 2022). Paired specimens of tumors and proximal peripheral tissues were examined by paired t tests to elucidate mean differences in gene expression between normal and cancerous tissues.

## 3. Results

### 3.1. Expression of COX1, COX2, ALOX5 and ALOX5AP in Colorectal Cancer

Violin density plots of the binary logarithmic expression levels of COX1, COX2, ALOX5 and ALOX5AP for the 411 cases without known MMR mutations are shown in Figure 1. All distributions were unimodal and approximately normal. Density plots for cases with MMR mutations and for male and female cases were also examined but did not differ significantly from the plots in Figure 1.

The mean expression of each inflammatory gene (COX1, COX2, ALOX5 and ALOX5AP) ranked above the 80th percentile of all genes. The mean expression level of COX1 exceeded COX2 (8.88 vs. 7.97 in binary log units) which translates to 2^0.91^ = 1.88 times higher mean expression of COX1 than COX2 (*p* < 0.01). Mean expression levels of ALOX5 and ALOX5AP were similar to one another (8.94 and 8.29 binary log binary units, respectively). The mean total COX expression (COX1 + COX2) was 16.8 binary log units, ranking in the upper 99th percentile of all levels of gene expression. Mean values of the inflammatory genes did not differ significantly by MMR status or gender.

### 3.2. Natural Antisense Transcript RP11-67C2.2

Figure 1 includes the distribution of RP11-67C2.2, a natural antisense transcript (NAT) that was co-expressed with ALOX5. In fact, the highest ALOX5 gene correlate observed was with RP11-67C2.2 (r = 0.78), a novel discovery in colorectal cancer. The RP11-67C2.2 gene is a long non-coding sequence of mRNA that is 462 base pairs in length. It is a unique Natural Antisense Transcript (NAT) that is homologous within intron 13 near the terminal 3′ end of the much larger ALOX5 gene (71.9 kilobases) on chromosome 10 at q11.21. The mean expression of RP11-C672.2 (2.94 binary log units) was approximately 1/64th of ALOX5 expression (8.94 units).

### 3.3. Correlations of COX1, COX2 ALOX5 and ALOX5AP in Colorectal Cancer

Correlations of the expression levels of inflammatory genes, COX1, COX2, ALOX5 and ALOX5AP are shown for the 411 colorectal cancer specimens without known MMR mutations in Table 1. Expression levels of ALOX5 were highly correlated with its nuclear membrane activating protein, ALOX5AP (r = 0.69) and ALOX5AP expression was also highly correlated with COX1 (r = 0.71). Correlations involving COX2 were of lesser magnitude, and in particular, COX2 was not highly correlated with COX1 (r = 0.47) or ALOX5 (r = 0.38).

### 3.4. Genes of Tyrosine Kinase Pathways: Src and JAK STAT

Hyperexpression of the Src and JAK STAT signaling cascades is known to increase survival, angiogenesis, proliferation and invasion of cancer cells in colorectal tumors and several other forms of malignancy [36].

We observed distinct clusters of correlations of inflammatory genes with genes of the Src and JAK STAT signaling pathways. As shown in Table 2, expression levels of ALOX5, ALOX5AP and COX1 were highly correlated with genes of the Src pathway (HCK, FGR, BTK and LAPTM5), whereas COX2 levels were highly correlated with genes of the JAK STAT pathway (CXCL8, CXCL5. IL6 and IL1B). Reciprocally, ALOX5, ALOX5AP and COX2 were not highly correlated with genes of JAK STAT, and COX2 was not highly correlated with genes of Src. We also observed that the expression levels of the genes of Src were highly correlated, as were the genes of JAK STAT, suggesting that the genes of these respective cell-signaling cascades were being expressed en bloc.

### 3.5. Expression of Src and JAK STAT Genes in Colorectal Cancer

Violin density plots of the binary logarithmic expression levels of genes of the Src and JAK STAT signaling pathways that were found to be highly correlated with inflammatory genes are shown in Figure 2a,b. The mean level of expression for the inflammation-correlated genes of the Src and JAK STAT signaling pathways ranged from 6.1 to 11.7 binary log units, each above the 75th percentile of all genes. Mean levels of these genes were similar in terms of MMR status and gender.

### 3.6. Epidermal Growth Factor Receptor (EGFR)

High expression of the well-known receptor tyrosine kinase, EGFR, triggers multiple cellular cascades such as MAPK and PI3K in colon carcinogenesis [42].

Figure 3 shows violin density plots for receptor tyrosine kinase genes (EGFR, TEK, OSMR) and other genes with tumor promoting activity (ATP8B2, DAAM1, SLAMF8, SRGN) as well as tumor suppressing activity (A2M, DAPK1, RASSF2 and SOCS3) that were highly correlated (r > 0.70) with one or more inflammatory genes.

Expression levels of EGFR were highly correlated with several genes of the Transforming Growth Factor (TGF-β) family and the Mitogen Activated Protein Kinase (MAPK) signaling cascade, e.g., the mean EGFR correlation with TGF-β1, TGF-β2 and TGF-β3 was r = 0.75 and correlations with MAPK1, MAP3K1, MAP3K2, MAPK3K3, MAP3K4 and MAP3K5 had a mean value of r = 0.79. Among the inflammatory genes, COX1 expression was highly correlated with MAP3K3 (r = 0.71) and TGF-β3 (r = 0.70).

In multivariate regression analysis, we found that expression levels of COX1, COX2, TGF-β3 and MAP3K3 explained more than 68% of EGFR variability (r = 0.82). The mean expression of EGFR was extremely high (10.7 binary log units), ranking above the 90th percentile of all genes among the 411 specimens of colon cancer without MMR mutations in TCGA. Observed mean values for other response genes depicted (TEK, OSMR, ATP8B2, DAAM1, RASSF2, DAAM1, SLAMF8, SRGN, A2M, DAPK, RASSF8 and SOCS3) were also high, ranking above the 95th percentile of all genes in TCGA.

### 3.7. Carcinoembryonic Antigen Cell Adhesion Molecules (CEACAM)

Expression levels of CEACAM genes (CEACAM1, CEACAM5, CEACAM6) were tightly correlated with one another, mean r = 0.77. Mean values were high ranging from 12.5 to 16.1 with an overall mean of 14,4 binary log units. Levels of the inflammatory genes, COX1, COX2, ALOX5 and ALOX5 were not highly correlated with expression of these CEACAM genes, mean r = 0.27, suggesting only a slight positive association between the inflammatory genes and expression of the CEACAM genes.

### 3.8. Mismatch Repair (MMR) Genes

Expression levels of MMR genes that are mutated in the Lynch Syndrome (MLH1, MSH2, MSH6, and PMS2) were tightly correlated with one another among specimens from colorectal cancer cases, both with and without MMR mutations, mean r = 0.73. These genes were highly expressed with mean values ranging from 8.0 to 10.5 with an overall mean of 9.5 binary log units. The inflammatory genes, COX1, COX2, ALOX5, and ALOX5 were correlated to a lesser extent with expression of the MMR genes, mean r = 0.53, suggesting a modest overall positive association.

### 3.9. Arachidonic Acid Biosynthesis and Release

Since arachidonic acid is the chief fatty acid used in the biosynthesis of prostaglandins and leukotrienes, correlations of COX1, COX2, ALOX5 and ALOX5AP levels of expression were specifically examined with genes encoding phospholipase enzymes (PLA2) and long-chain acyl-coenzyme A synthetases (ACSL) that modulate arachidonic acid biosynthesis and release [2,3,4,15]. Notably, these genes were expressed at high levels in the dataset, most ranking above the 95th percentile, e.g., ACSL1, ACSL3, ACSL4, and ACSL5 (overall mean = 11.2 binary log units) and PLA2G2A, PLA2G4A, PLA2G5 and PLA2G7 (overall mean = 8.8 units).

Correlations of inflammatory genes with ACSL and PLA2 levels were positive but individual correlations were only moderately elevated (overall mean r = 0.44, r^2^ = 19%). The highest of these correlations was between the expression of ALOX5AP and PLA2G7 (r = 0.73). The highest correlations involving ACLS genes were between COX1 and COX2 levels with ACLS4 (mean r = 0.51).

### 3.10. Cell Membrane Receptors for Prostaglandins and Leukotrienes

Prostaglandins and leukotrienes exert their effects in the colorectal tumor microenvironment through interactions with specific cell membrane receptors of immune cells and cancer cells [44,45]. We therefore examined correlations between COX1 and COX2 expression and genes encoding cell membrane receptors for the major inflammatory prostaglandin, PGE2 (PTGER1, PTGER2, PTGER3, PTGER4 and PTGFR), and between ALOX5 and ALOX5AP with cell membrane receptors for the major inflammatory leukotriene, LTB4 (LTB4R) and cysteinyl leukotrienes, CysLT (CysLTR), as well as with Colony Stimulating Factor Receptors (CSFR) that have been implicated in immune cell signaling [46,47,48].

Mean expression levels of genes encoding PGE2 receptors (PTGER1, PTGER2, PTGER3, PTGER4 and PTGFR) varied widely ranging from 3.6 to 9.6 with an overall mean of 6.4 binary log units. Expression of COX1 was highly correlated with PTGER3 (r = 0.62) and PTGFR (r = 0.68) whereas other correlations of COX1 and COX2 with PGE2 cell membrane receptors were positive but of lesser magnitude.

Expression levels of COX1, ALOX5 and ALOX5AP were all highly correlated with colony stimulating factor receptors, CSFR1, CSFR2B and CSFR3 (mean r = 0.68), and the ALOX5AP correlations were particularly high (mean r = 0.81). Expression levels of ALOX5AP were also highly correlated with the cysteinyl leukotriene receptors, CysLTR1, CysLTR2 (mean r = 0.68). The observed correlations of ALOX5 and ALOX5AP with LTB4R levels were positive but of lesser magnitude (mean r = 0.23). Mean expression levels of genes encoding LTB4R and the CSF receptors ranged from 7.6 to 9.7 with an overall mean of 8.5 binary log units. Genes encoding the cysteinyl leukotriene receptors were expressed at lower levels (overall mean = 4.5 units).

### 3.11. Genetic Expression in Paired Samples

Paired samples of benign tissues proximal to tumor specimens were available for 45 of the 469 colorectal samples in TCGA. Mean expression levels of COX1, COX2, ALOX5, ALOX5AP and correlated genes of the JAK STAT and Src pathways in adjacent tissues were similar to levels in the tumor samples.

## 4. Discussion

Our results confirm that COX1, COX2, ALOX5 and ALOX5AP, rate-limiting enzymes of prostaglandin and leukotriene biosynthesis, are all constitutively expressed in colorectal cancer. Distributions of mRNA values in binary log units were unimodal with mean values above the 80th percentile for each of the four inflammatory genes. Notably, expression levels of ALOX5 and particularly ALOX5AP were highly correlated with COX1 but not COX2. Furthermore, the expression levels of COX1, ALOX5 and ALOX5AP were highly correlated with major genes of the Src pathway (HCK, FGR, BTK and LAPTM5), whereas COX2 expression was highly correlated with major genes of the JAK STAT pathway (IL6, ILB1, CXCL5 and CXCL8).

A novel finding is that a Natural Antisense Transcript (NAT), RP11-C672.2, had the highest correlation (r = 0.78) with ALOX5 expression. The gene is a unique long non-coding mRNA located near the 3′ terminal region of ALOX5 on chromosome 10 (10q11.21). Such genes have been found to modulate the expression of their parent genes, either promoting or inhibiting expression. The positive nature of the observed correlation suggests that RP11-C672.2 is a conserved natural antisense transcript (NAT) that enhances expression of ALOX5 during tumorigenesis. One possible mechanism of action is interference with feedback inhibition of ALOX5 expression due to homology with miRNA [49]. This finding clearly needs further investigation, as RP11-C672.2 may be a target for gene therapy of colorectal cancer or other inflammatory conditions.

Several other genes that impact colorectal cancer development were also highly correlated with the COX and LOX genes. These included well-known receptor tyrosine kinase genes (EGFR, TEK and OSMR) that activate MAPK and PI3K signaling pathways [40], and other tumor-promoter genes (ATP8B2, DAAM1, SLAMF8 and SRGN) [50,51,52,53] as well as tumor-suppressor genes (A2M, DAPK1, RASSF8 and SOCS3) [54,55,56,57]. Our results suggest that both the prostaglandin and leukotriene inflammatory cascades not only independently contribute to colorectal cancer development, but also that there is substantial crosstalk between the COX and LOX inflammatory cascades in colorectal carcinogenesis.

### 4.1. Inflammogenesis Model of Colorectal Cancer

Figure 4 shows a model of inflammogenesis of colorectal cancer based on the results of our investigation. The model postulates that colorectal carcinogenesis involves sustained constitutive overexpression of COX1, COX2, ALOX5 and ALOX5AP and chronic inflammation leading to malignant transformation of the epithelium of the colon and rectum. It includes only the highest correlates of COX1, COX2 ALOX5 and ALOX5AP. The model is speculative since it is based on the assumption that gene expression levels quantified by mRNA accurately represent levels of relative protein synthesis and gene function.

The tumor microenvironment obviously involves the expression of genes in epithelial-derived cancer cells plus several other types of cells, e.g., adipocytes, macrophages, granulocytes, lymphocytes and myeloid-derived stem cells. Specific functions of genes in the model are consistent with amplified epithelial–mesenchymal transition, mitogenesis, mutagenesis, angiogenesis, cell survival/dysregulated apoptosis, and metastasis in response to the overexpression of COX1, COX2, ALOX5 and ALOX5AP in the inflammogenesis of colorectal cancer.

In the model, biosynthesis and release of arachidonic acid are catalyzed from cell membrane phospholipids by phospholipase A2 (PLA2) and acyl-CoA synthetases (ACSL) [58]. The chief inflammatory enzymes, cyclooxygenase 1, 2 (COX1, COX2) catalyze prostaglandin biosynthesis, and arachidonate lipoxygenase-5 and its activating protein (ALOX5, ALOX5AP) catalyze leukotriene biosynthesis from arachidonic acid. The novel natural antisense transcript RP11-67C2.2 may help upregulate ALOX5. Inflammatory prostaglandins and leukotrienes circulate in the extracellular milieu and activate cell membrane receptors of macrophages, lymphocytes, and other immune cells (PTGE1, PTGE2, PTGE3, PTGE4, PTGFR, LTB4R, CysLTR1, CysLTR2, CSF1R, CSF2RB, CSF3R). Cytokines, ligands and growth factors secreted primarily by immune cells (IL6, IL1B, CXCL5, CXCL8, TGF-a, TGF-β) activate tyrosine kinase receptors (RTK) and signaling pathways of epithelial cells (JAK STAT, EGFR, TEK, OSMR, MAPK, PI3K, Src) involving phosphorylation of intracellular proteins (BTK, FGR, HCK, LAPTM5) that incite transcription of specific nuclear genes (ATP8B2, DAAM1, SLAMF8, SRGN) that promote colorectal tumorigenesis.

Tumor suppressor genes that counteract tumorigenesis are also in the model. These include mismatch repair genes (PMS2, MLH1, MLH5, MLH6), genes encoding cell adhesion molecules (CEACAM1, CEACAM5, CEACAM6), suppression of cytokine signaling (SOCS3), cell necrosis (DAPK1), protease inhibition (A2M), and blockade of cell-signaling cascades (RASSF8).

### 4.2. Gene Targets for Prevention and Therapy of Colorectal Cancer

The proposed inflammogenesis model of colorectal cancer (Figure 4) includes several putative ‘gene targets’ for prevention and therapy. Specifically, selective tissue-specific inhibition of the chief rate-limiting enzymes of the prostaglandin and leukotriene cascades encoded by COX1, COX2, ALOX5, ALOX5AP as well as the novel NAT gene, RP11-C672.2, may be effective in turning off multiple downstream tumor-promoting response genes of the model. Implicit in the model is that multiple factors and cell types are under significant control of the human inflammatory cascades in the genesis of colorectal cancer. As has been pointed out by other investigators, simultaneous inhibition of both the COX and LOX cascades would appear to be a more plausible approach to reduce the progression of colorectal cancer than only blocking one cascade. This approach has been effectively applied in animal models of carcinogenesis and is now being tested in humans for both chemoprevention and therapy [16,22,25].

### 4.3. Conserved Cell-Signaling Cascades in Carcinogenesis

In a previous report, we examined the genetic expression of COX1, COX2, ALOX5 and ALOX5AP in breast cancer specimens from TCGA [59]. A surprising finding was that the estrogen synthetase, aromatase, was co-expressed with the inflammatory genes across all subtypes of breast cancer. Nevertheless, similar to our results presented here for colorectal cancer, we found that key genes of major cell-signaling cascades that drive tumorigenesis, e.g., JAK STAT (IL6), Src, RTK (EGFR), PI3K, and MAPK, were also highly correlated with the expression of inflammatory genes in breast cancer. Results suggest that inflammatory cell-signaling pathways are remarkably conserved in the development of these two major forms of malignancy and possibly others. Future studies are needed to elucidate the value of simultaneous blockade of these inflammatory gene targets in cancer prevention and therapy.

### 4.4. Limitations of mRNA Data

Our analysis used mRNA seq as a measure of gene expression in colorectal cancer specimens from TCGA. Protein synthesis and function involves a complex regimented sequence of events including nuclear transcription of DNA to mRNA, elongation and transport of mRNA to the cytoplasm, ribosomal translation by tRNA and folding, modification and assembly of amino acid sequences in the endoplasmic reticulum, further packaging and transport by the Golgi apparatus, and ultimate biodegradation. The level of mRNA is therefore not necessarily a valid measure of gene function, and several studies have observed only weak associations between mRNA and protein synthesis, particularly for non-differentiated genes that are not specific for disease processes [60,61,62]. Nevertheless, recent reports suggest that for differentiated genes that are associated with the pathogenesis of cancer, mRNA levels encoded by tumor-promoting genes may in fact be valid measures of relative protein synthesis and gene [63,64]. However, the absence of proteomic measurement is a limitation of our findings and should be examined in future studies to provide additional validation that the observed high expression of mRNA by genes that regulate inflammation and correlated tumor-promoting genes play an important role in colorectal cancer development.

## 5. Conclusions

Key findings from our analysis of gene-specific mRNA expression in colorectal cancer tissues in The Cancer Genome Atlas (TCGA) dataset include the following: (1) COX1, COX2, ALOX5 and ALOX5AP are constitutively over-expressed by multiple cell types in the tumor microenvironment (adipose, macrophages, myeloid-derived suppressor cells, lymphocytes and malignant cells); (2) genes of multiple cell-signaling cascades that promote colorectal carcinogenesis (EGFR, OSMR, TEK, Src, JAK STAT MAPK and PI3K) are highly co-expressed with the COX and LOX genes, and (3) essential features of colorectal carcinogenesis (epithelial to mesenchymal transition, mutagenesis, mitogenesis, angiogenesis, cell survival/dysfunctional apoptosis, immunosuppression and metastasis) are linked to COX1 and COX2-driven prostaglandin E2 (PGE-2) biosynthesis and ALOX5 and ALOX5AP-driven leukotriene biosynthesis with the correlated over-expression of multiple tumor-modulating response genes.

A novel finding is the co-expression of a novel natural antisense transcript, RP11-C672.2, with the ALOX5 gene. Additional molecular studies are needed to determine if this long noncoding natural antisense transcript is involved in the upregulation of ALOX5.

Cohesive scientific evidence from molecular, animal, and human investigations supports the hypothesis that sustained constitutive overexpression of COX and LOX is a ubiquitous driver of colorectal carcinogenesis, and reciprocally, that blockade of these major inflammatory cascades has strong potential for colorectal cancer prevention and therapy. The current study adds to the evidence that COX and LOX overexpression is irrevocably linked to colorectal cancer development. The totality of evidence clearly supports the supposition that colorectal carcinogenesis often evolves as a progressive series of highly specific cellular and molecular changes in response to induction of constitutive overexpression of COX and LOX genes and their respective prostaglandins and leukotrienes in the “inflammogenesis of colorectal cancer”.

## Figures and Tables

**Figure 1 cancers-15-02380-f001:**
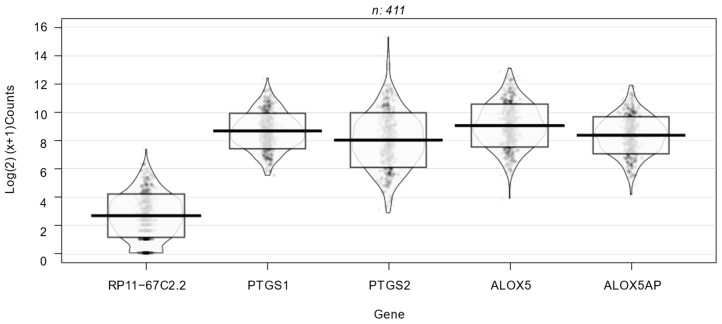
Violin density plots of the genetic expression of COX1, COX2, ALOX5 and ALOX5AP in invasive colorectal cancer. Means and standard deviations are indicated by horizontal bars and box plots, respectively. Prostaglandin Synthetase-1, PTSG1 (Cyclooxygenase-1, COX1). Prostaglandin Synthetase-2, PTSG2 (Cyclooxygenase-2, COX2). Arachidonate 5-Lipoxygenase (ALOX5). Arachidonate 5-Lipoxygenase Activating Protein (ALOX5AP). RP11-67C2.2: Novel Natural Antisense Transcript Gene, RP11-67C2.2 nested within the terminal region of the ALOX5 gene on chromosome 10q11.21.

**Figure 2 cancers-15-02380-f002:**
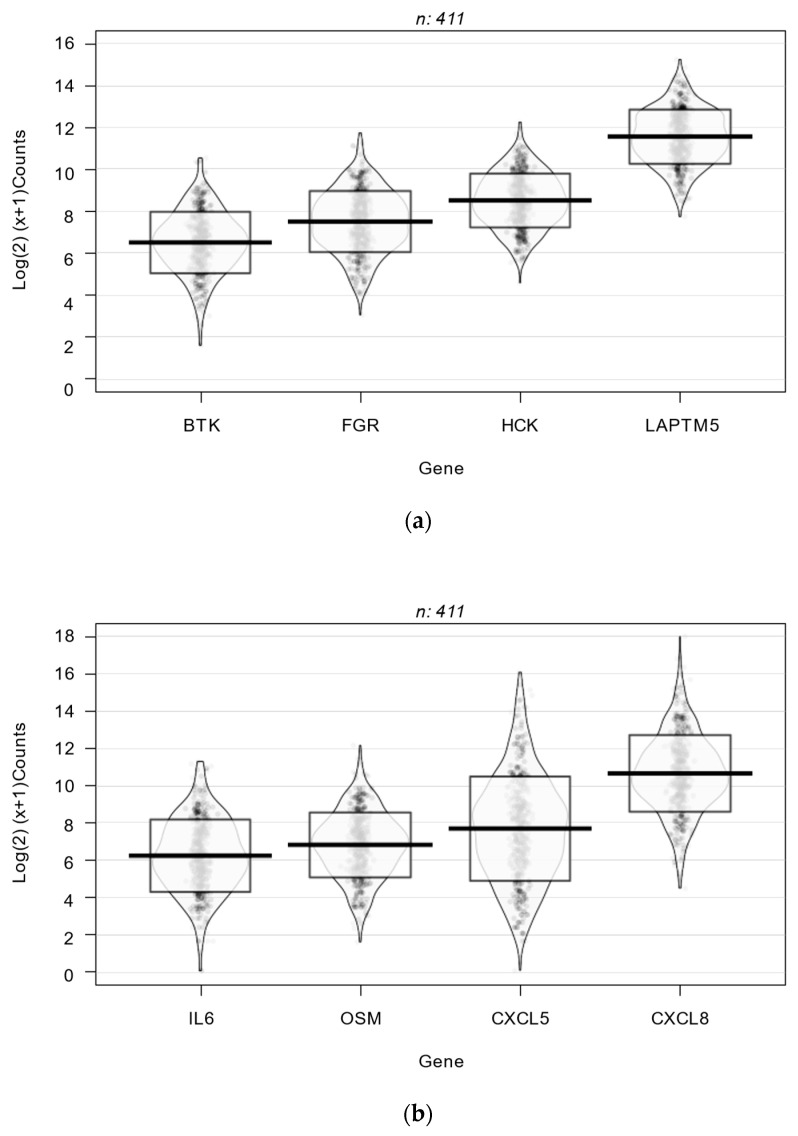
Violin density plots of the expression of inflammation-correlated genes of the Src and JAK STAT signaling pathways. (**a**) Src Genes; (**b**) JAK STAT Genes.

**Figure 3 cancers-15-02380-f003:**
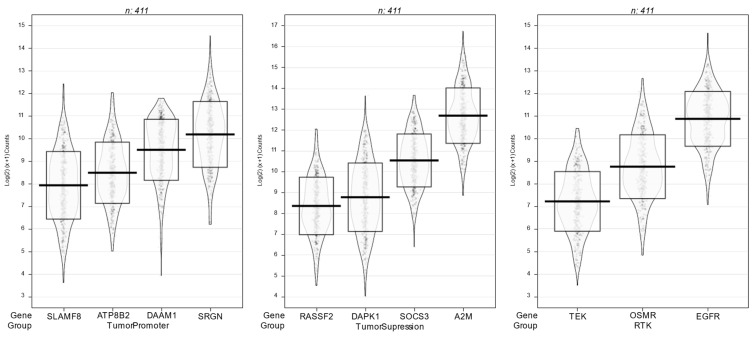
Violin density plots of the expression of EGFR and other inflammation-correlated tumor response genes.

**Figure 4 cancers-15-02380-f004:**
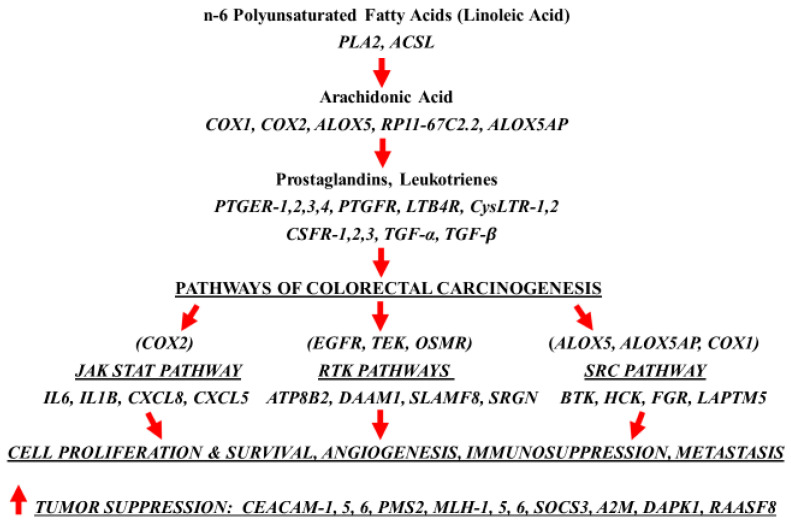
Inflammogenesis model of colorectal carcinogenesis. Genes and pathways in the model. Arachidonic Acid biosynthesis: Phospholipase A2: PLA2, Acyl-CoA synthetase: ACSLl; Prostaglandin biosynthesis: Cyclooxygenase-1, 2 (COX1, COX2), Leukotriene biosynthesis: ALOX5 arachidonate 5-lipoxygenase, RP 11 67C2.2 long noncoding natural antisense transcript of ALOX5, ALOX5AP (FLAP) arachidonate 5-lipoxygenase activating protein; Prostaglandin E receptors: PTGER1, PTGER2, PTGER3, PTGER4, PTGFR; Leukotriene receptors: LTB4R, CysLTR1, CysLTR2; Colony Stimulating Factor receptors: CSFR1, CSFR2B, CSFR3; Transforming Growth Factor: TGF; Janus Kinase Transduction and Transcription cell signaling: JAK STAT; Receptor Tyrosine Kinase: RTK; Epidermal Growth Factor Receptor: EGFR; Mitogen Activated Protein Kinase cell signaling: MAPK; Phosphoinositide 3 Kinase cell signaling: PI3K; Src Tyrosine Kinase cell signaling: Src; TEK Receptor Tyrosine Kinase: TEK; Oncostatin Tyrosine Kinase Membrane Receptor: OSMR; Interleukin 6: IL6; Interleukin 1B: IL1B; CXC Motif Chemokine Ligands 5 and 8: CXCL5, CXCL8; ATPase Phospholipid Transporting 8B2: ATP8B2; Disheveled Activator of Morphogenesis 1: DAAM1; Signaling Lymphocytic Activation Molecule Family 8: SLAMF8; Serglycin: SRGN; Bruton’s Tyrosine Kinase: BTK; Feline Gardner–Rasheed proto-oncogene: FGR; Hematopoietic Cytokinase: HCK; Lysosomal Activating Transmembrane Protein-5: LAPTM5; Carcinoembryonic antigen-related cell adhesion molecules: CEACAM 1, 5, 6; Mismatch Repair Genes: PMS2, MLH 1, 5, 6; Suppressor of Cytokine Signaling: SOCS3; Alpha 2 Macroglobulin (Protease Inhibitor): A2M; Death Associated Protein Kinase: DAPK1; Ras Associated Family: RAASF8.

**Table 1 cancers-15-02380-t001:** Pearson correlation coefficients for COX1, COX2, ALOX5 and ALOX5AP among 411 specimens of invasive colorectal cancer.

GENE	COX2	ALOX5	ALOX5AP
COX1	0.47	0.55	0.69
COX2	---	0.38	0.50
ALOX5		---	0.64

**Table 2 cancers-15-02380-t002:** Correlations of ALOX5, ALOX5AP, COX1 and COX2 with Src and JAK STAT genes in colorectal cancer.

PATHWAY and GENES	Genes of Src Pathway				Genes of JAK STAT Pathway			
	HCK	FGR	BTK	LAPTM5	CXCL8	IL6	IL1B	CXCL5
ALOX5	0.65	0.62	0.58	0.64	0.29	0.27	0.37	0.14
ALOX5AP	0.88	0.88	0.80	0.89	0.58	0.53	0.53	0.42
COX1	0.73	0.68	0.68	0.69	0.34	0.44	0.44	0.32
COX2	0.46	0.45	0.46	0.46	0.75	0.73	0.72	0.65

## Data Availability

Results are based on data generated by The Cancer Genome Atlas (TCGA) managed by the National Cancer Institute (NCI) and the National Human Genome Research institute (NHGRI), http://genome.nih.gov, accessed on 10 November 2022.

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
