# Peer review of "Cyclooxygenase and Lipoxygenase Gene Expression in the Inflammogenesis of Colorectal Cancer: Correlated Expression of EGFR, JAK STAT and Src Genes, and a Natural Antisense Transcript, RP11-C67.2.2"

_cancers, 2023, doi:10.3390/cancers15082380_

Round 1

Reviewer 1 Report

In this manuscript entitled „Cyclooxygenase and Lipoxygenase Gene Expression in The Inflammogenesis of Colorectal Cancer: Correlated Expression of EGFR, JAK STAT and Src Genes, and a Natural Antisense Transcript, RP11-C67.2.2” Kennedy B. M. and Harris R. E. focused on relationship between the expression of major genes that regulate inflammation in colorectal cancer (cyclooxygenase-1, 2, arachidonate-5-lipoxygenase, arachidonate-5-lipoxygenase activating protein) and expression of genes of major cell signaling pathways involved in tumorgenesis (Src, JAK STAT, MAPK, PI3K, and especially a natural antisense transcript, RP11-67C2.2, a long non-coding mRNA that is located within the terminal intron of the ALOX5 gene. The role of inflammation is tightly link to the risk of many forms of cancer. It is a very interesting topic of the last decade. This manuscript was written well. The interpretation of the data was done correctly.

However, I have concern about interpretation of the mRNA levels from data retrieved only from TCGA database. The authors should support their conclusion with different transcriptional/proteomic analysis. It would be needed to validate the data from CTGA by real-time PCR. The examination of protein expression showing by another methods (e.g. western blot, immunohistochemistry or immunofluorescence) is also needed.

Author Response

Reviewer One

  1. We have added a paragraph on the limitations of the mRNA data analyzed from TCGA with key references from the recent peer reviewed literature (4.4 in the text). We point out that for genes not specifically related to disease processes such as tumorigenesis, correlations of mRNA with protein levels are relatively weak.  Nevertheless, recent studies suggest that for genes specifically involved in carcinogenesis, mRNA expression  is tightly linked to protein biosynthesis and gene function.  References 60-65 have been added to support these statements. Our paragraph stresses that more studies are needed to explore linkages between mRNA, protein biosynthesis and protein function for genes that regulate inflammation and correlated tumor-promoting genes.

Reviewer 2 Report

Authors have analyzed data on colorectal cancers from the open access tier of TCGA, and identify several genes that are overexpressed on such cancers when compared to normal tissues.

The obtained overexpresion data is sound but the conclusion drawn to built the general tumorigenesis model is highly speculative.

In the introduction references backing up the imflamogenesis of cancer are not recent 9,11,20,21 and 

references 24,25,26,27; back up licofelone but not curcumin activity against cancer.

Author Response

Reviewer Two

  1. We have substituted a reference documenting the inhibitory impact of curcumin on the ALOX5/leukotriene cascade (Ref 26).
  2. We would like to retain historical references cited on the role of inflammation in the genesis of colorectal cancer and other malignancies.
  3. We have added a statement about the speculative nature of the proposed model of colorectal cancer that is based on the assumption that mRNA seq data is linked to protein biosynthesis and function.

Round 2

Reviewer 1 Report

You had better add the "wet" methods. The research would be more complete. On the other hand, tissues are hard to obtain and manuscript is well-written.